# Analysis of low-level somatic mosaicism reveals stage and tissue-specific mutational features in human development

Ja Hye Kim[1]☯, Shinwon Hwang[2,3]☯, Hyeonju Son[2,4], Dongsun Kim[2,4], Il Bin Kim[1,5], Myeong-Heui Kim[1,6], Nam Suk Sim[1,7], Dong Seok Kim[8], Yoo-Jin Ha[2,4], Junehawk Lee[9], Hoon-Chul Kang[10], Jeong Ho Lee[1,6]*, Sangwoo Kim[2,4]*

1 Graduate School of Medical Science and Engineering, Korea Advanced Institute of Science and Technology (KAIST), Daejeon, Republic of Korea, 2 Department of Biomedical Systems Informatics, Yonsei University College of Medicine, Seoul, Republic of Korea, 3 Department of Medicine, Physician-Scientist Program, Yonsei University College of Medicine, Seoul, Republic of Korea, 4 Graduate School of Medical Science, Brain Korea 21 Project, Yonsei University College of Medicine, Seoul, Republic of Korea, 5 Department of Psychiatry, Hanyang University Guri Hospital, Guri, Republic of Korea, 6 SoVarGen Inc., Daejeon, Republic of Korea, 7 Department of Otorhinolaryngology, Yonsei University College of Medicine, Seoul, Republic of Korea, 8 Department of Neurosurgery, Pediatric Neurosurgery, Severance Children's Hospital, Yonsei University College of Medicine, Seoul, Republic of Korea, 9 Center for Supercomputing Applications, National Institute of Supercomputing and Networking, Korea Institute of Science and Technology Information, Daejeon, Republic of Korea, 10 Division of Pediatric Neurology, Department of Pediatrics, Pediatric Epilepsy Clinics, Severance Children's Hospital, Epilepsy Research Institute, Yonsei University College of Medicine, Seoul, Republic of Korea

☯ These authors contributed equally to this work.
* jhlee4246@kaist.ac.kr (J.H.L.); swkim@yuhs.ac (S.K.)

**Data Availability Statement:** All datasets generated and/or analyzed during the study are publicly available in NCBI Sequence Read Archive (SRA) with the accession numbers: PRJNA276391

## Abstract

Most somatic mutations that arise during normal development are present at low levels in single or multiple tissues depending on the developmental stage and affected organs. However, the effect of human developmental stages or mutations of different organs on the features of somatic mutations is still unclear. Here, we performed a systemic and comprehensive analysis of low-level somatic mutations using deep whole-exome sequencing (average read depth ~500×) of 498 multiple organ tissues with matched controls from 190 individuals. Our results showed that early clone-forming mutations shared between multiple organs were lower in number but showed higher allele frequencies than late clone-forming mutations [0.54 vs. 5.83 variants per individual; 6.17% vs. 1.5% variant allele frequency (VAF)] along with less nonsynonymous mutations and lower functional impacts. Additionally, early and late clone-forming mutations had unique mutational signatures that were distinct from mutations that originated from tumors. Compared with early clone-forming mutations that showed a clock-like signature across all organs or tissues studied, late clone-forming mutations showed organ, tissue, and cell-type specificity in the mutation counts, VAFs, and mutational signatures. In particular, analysis of brain somatic mutations showed a bimodal occurrence and temporal-lobe-specific signature. These findings provide new insights into the features of somatic mosaicism that are dependent on developmental stage and brain regions.

(Focal Cortical Dysplasia), PRJNA470641 (Glioblastoma), PRJNA481075 (Ganglioglioma), PRJNA532465 (Alzheimer's Disease and non-Alzheimer's dementia), PRJNA604458 (Schizophrenia), PRJNA607437 (Autism Spectrum Disorder), PRJNA868096 (Major Depressive Disorder, LumboSacral lipoma, Non-Syndromic Craniosynostosis, Non-Lesional Epilepsy).

**Funding:** This research was supported by the National Research Foundation of Korea (NRF) grant funded by the Korea government (MSIT) (No. 2019R1A2C2008050 to S. K.), Team Science Award of Yonsei University College of Medicine (grant/award number: 6-2021-0007 to S. K.), the Suh Kyungbae Foundation (to J.H.L.), a National Research Foundation of Korea (NRF) grant funded by the Korean Ministry of Science and Information and Communication Technology (ICT) (No. 2019R1A3B2066619 to J.H.L), the Korea Health Technology R&D Project through the Korea Health Industry Development Institute (KHIDI) and Korea Dementia Research Center (KDRC), funded by the Ministry of Health & Welfare and Ministry of Science and ICT, Republic of Korea (grant number : HU21C0286). The funders had no role in study design, data collection and analysis, decision to publish, or preparation of the manuscript.

**Competing interests:** I have read the journal's policy and the authors of this manuscript have the following competing interests: J.H.L. is the co-founder and CTO of SoVarGen Inc., which seeks to develop new diagnostics and therapeutics for brain disorders. The other authors declare no competing interests.

## Author summary

Most somatic mutations that arise during normal development are present at low levels in single or multiple tissues, and often show a degree of clonality depending on the time and origin of the mutation. Recent studies have identified the characteristics of postzygotic variants of somatic mutations at the single-cell or mono-clonal levels. However, the results may not be fully representative of the mutational processes involved. Here, we describe a comprehensive analysis of low-level somatic mutations identified after deep whole-exome sequencing in peripheral and brain tissues. We found that clone-forming mutations are uniquely defined by early and late-stage aspects in the mutational profiles. Thus, we identified reliable spatiotemporal characteristics of mosaic variants. Additionally, we found low-level clone-forming mosaic variants across multiple stages and tissues, and identified their intrinsic features.

## Introduction

Somatic mutations persistently occur in normal cells during the entire human lifetime [1]. Although unaccompanied with unregulated proliferation as occurs in cancer, the degree of clonality of somatic mutations often depends on their time of occurrence and origin. For example, variants in the early stages of development tend to affect multiple organs in different germ layers and show high variant allele frequencies (VAFs), whereas those in later stages have low VAFs [2, 3]. Somatic variants that occur after birth are potentially transient and restricted to the cellular level. However, mutations in stem or progenitor cells [4], or variants that confer clonal expansion [5], are persistent and accumulate during the lifetime of an individual. Mutations in stem cells show a sufficient level of VAFs that can be detected via bulk-genome sequencing of tissues. Tissue-level somatic mutations are crucial for the pathogenicity of non-cancerous or benign diseases, and the aberration magnitude is associated with the allele frequency of the mutations [6, 7]. For example, mTOR-pathway-activating somatic mutations cause two types of intractable epilepsy (hemimegalencephaly and focal cortical dysplasia) depending on the time of occurrence of the mutation and VAFs (10%–30% VAFs in hemimegalencephaly, and 1%–10% of VAFs in focal cortical dysplasia) [8–10]. Advances in genetic technologies and increased sequencing efficiencies have helped identify characteristics of somatic mutations at the single-cell or mono-clonal levels [11–15]. However, the detectable variants identified using these methods may not be fully representative of the tissue types and mutational processes involved because of the technical challenges associated with single-cell sequencing (e.g., whole gene amplification), restricted cell types (e.g., stem/progenitor cell or reprogrammed cells), or culturing procedure used. The mutational profiles of clone-forming cells during development or aging can help identify tissue types, mutation timing, disease-specific patterns, and distributions in benign disease or normal individuals, despite limited explanations for cellular heterogeneity [16]. Despite several studies on somatic variants in the literature, it is unclear how low-level but clone-forming somatic mosaicisms are characterized by the time and location of occurrence due to the lack of multi-organ sequencing data and its analytical limitations.

To address these issues, we performed a comprehensive analysis of low-level somatic mutations identified using deep whole-exome sequencing (WES) data (average read depth: ~500×) from 498 tissues of 190 individuals. The cohorts analyzed consisted of two or more organs, including the brain and matched-peripheral tissues, and covered a variety of neurological

conditions. The pairwise analysis of age, brain locations, and diseases enabled multi-dimensional analysis and direct comparison with cancer mutations identified using the same analysis procedure.

## Results

### Three somatic mutation analysis categories including early-stage, late-stage, and tumor mutations

The cohorts consisted of multiple organs, including brain ($n = 301$), blood ($n = 100$), liver ($n = 60$), heart ($n = 13$), and other peripheral tissues ($n = 24$). The 190 individuals from whom tissues were obtained included patients with 'non-tumor' neurological disorders ($n = 133$), brain tumors (glioblastoma and ganglioglioma, $n = 19$), and non-diseased controls ($n = 38$) (Fig 1A and S1 Table).

We defined and used three different categories in the analysis: early-stage, late-stage, and tumor mutations (Fig 1B). Early-stage mutations were defined as shared mutations in multiple organs that may occur during early embryonic development before gastrulation. Late-stage mutations are restricted to a single organ and are more likely to be induced by clone-forming cells containing late embryonic (post-gastrulation) or post-natal somatic mutations, although a few mutations may be present during the pre-gastrulation stage. Tumor specimens were acquired from the primary or recurrent tumor, and microscopic tumor analysis via histological evaluation was performed by a neuropathologist using the World Health Organization (WHO) 2016 grading criteria [17].

Based on the above definitions, somatic mutation identification was performed using an ensemble of robust variant calling algorithms such as Mutect2 [18], RePlow [19], and NeuSomatic [20] for the 1034 potential sample pair combinations. After strict filtration and tests for organ specificity using one-sample proportion tests (Fig 1A and Materials and Methods), we detected 103 early and 997 late-stage mutations, as well as 583 tumor mutations. To validate the identified mutations, 114 randomly selected single nucleotide variants (SNVs; ~10% of non-tumor mutations) were sequenced using Sanger sequencing and via targeted amplicon sequencing (TASeq) to obtain an ultra-high depth (average: 507,856× reads). Our call set showed high accuracy for the identification of the early-stage (89.47%, 17/19) and late-stage and tumor mutations (90.24%, 74/82) (Fig 1C and S2 Table). A high concordance in VAFs across tissues (Pearson's correlation $r = 0.84$; $P = 1.00 \times 10^{-42}$) and between the WES and TASeq data ($r = 0.61$; $P = 1.17 \times 10^{-48}$) indicated the accuracy of the calls (Fig 1D).

Additionally, we compared the quantitative traits of the mutations including the number and allele frequency at different stages. On average, we identified 0.18 early and 2.8 late-stage somatic mutations per individual (Fig 2A). The number of mutations in normal tissues was substantially lower than that in tumors (8.12 per individual). The overall number of late-stage and tumor mutations was positively correlated with age (Pearson's $r$: late-stage, 0.45; and tumor, 0.36) (Fig 2B). However, the VAF of late-stage mutations was inversely correlated with age, probably due to the accumulation of late-occurring somatic mutations in blood cells with a low allelic fraction. Conversely, we found no correlation between early-stage mutations and age, indicating that these mutations are confined to early developmental stages. The VAFs of the mutations were higher in the early ($6.17 \pm 3.32\%$) relative to the late stage ($1.50 \pm 3.29\%$) of development (Figs 2C and S1A and S1B).

### Three distinct mutation spectra identified by mutational signature analysis

Next, we performed a mutation profiling analysis to investigate the underlying mutagenic processes in somatic cells (Fig 3A–3D). *De novo* signature extraction of the 1494 somatic SNVs

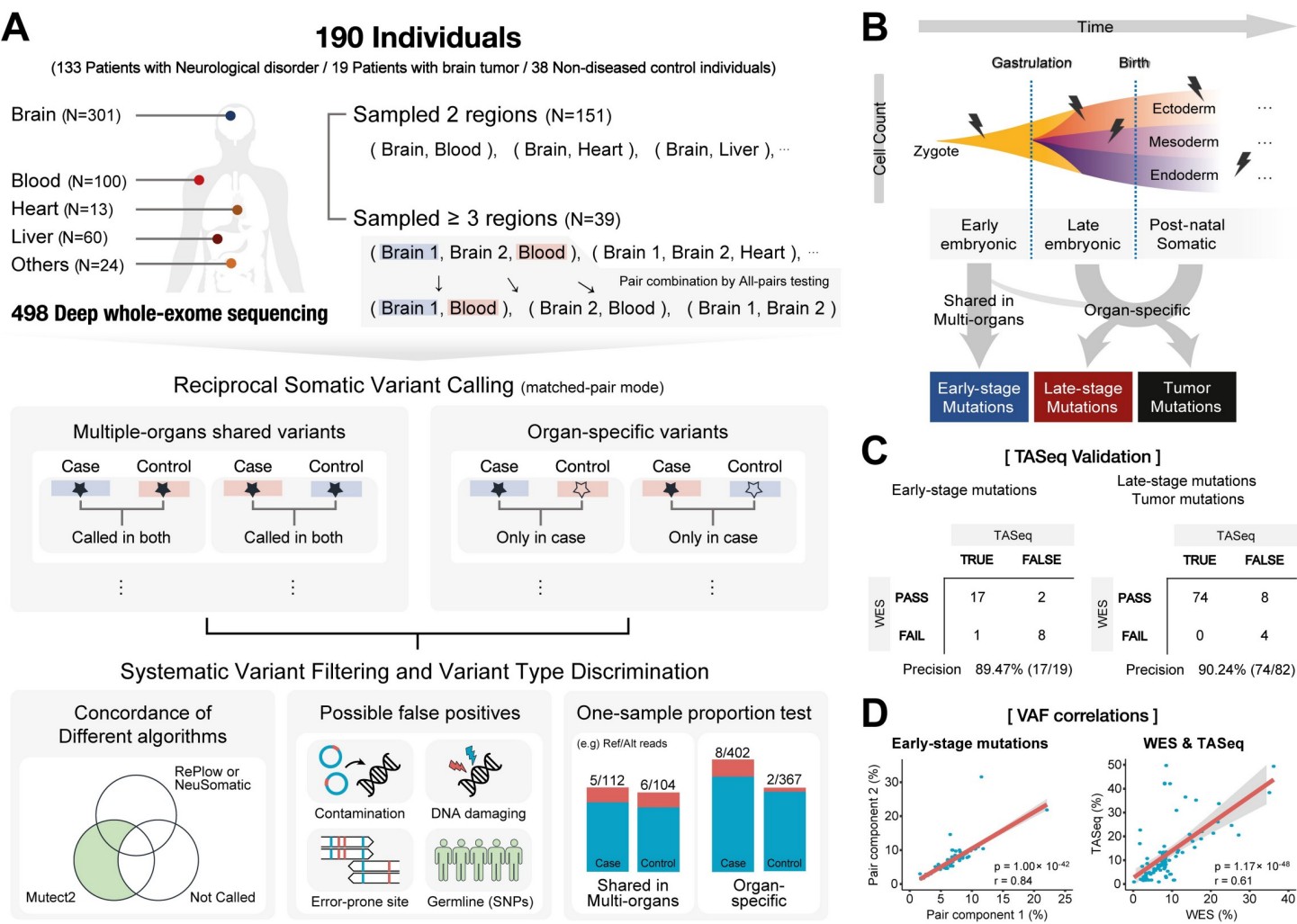

**Fig 1. Detection of early and late-stage somatic variants in the brain and matched peripheral tissues.** (**A**) Schematic flow showing the bioinformatics pipelines for the analysis of 301 brain tissues and 197 peripheral tissues from 190 individuals. Tumor mutations were identified from primary or recurrent tumor specimens, and confirmed by histologic examination. To identify somatic variants, Mutect2 and RePlow/NeuSomatic were used for reciprocal mutation calling by all-pairs testing. First, to detect somatic mutation in tissue "A", tissues "A" and "B" were used as a case and control, respectively, and tissue "A" was further used as a control to identify somatic mutations in tissue "B". For tumor tissues, the same analysis was performed as in other samples. Subsequently, post-call filtering was used. After strict filtration and tests for organ specificity using one-sample proportion tests, the shared and organ-specific mutations were classified. The colored bar indicates read distribution, and the red and blue-colored bars indicate the reference and altered reads, respectively (lower right panel). (**B**), (**C**) Early-stage, late-stage, and tumor mutations were classified with a highly accurate precision rate (89.47%, early-stage; and 90.24% in late-stage and tumor mutations). We determined organ specificity using a one-sample proportion test. (**D**) Correlation of VAFs from two matched tissues and the WES and TASeq data. VAFs were highly concordant between paired tissues ($r = 0.84$; $P < 0.0001$), and the WES and TASeq data ($r = 0.61$; $P < 0.0001$).

(94 early, 880 late stage, and 520 tumor SNVs) helped identify three novel signatures (Fig 3A), including signatures A, B1, and B2, all of which exhibited C>T as the major base substitution, and showed an additional T>C enrichment in signature A. Despite the overall similarity in the mutational spectrum, especially between B1 and B2 (cosine similarity: 0.95), a clear distinction was observed in the relative contribution of the mutations to the sample groups, indicating the uniqueness of the signatures (i.e., signatures A, B1, and B2 predominantly contributed to the early, late-stage, and tumor SNVs, respectively) (Fig 3B).

Mapping of the three signatures to COSMIC Mutational Signatures (v3.1; June 2020) [21] identified clock-like SNVs (SBS1, SBS5, and SBS40), and small insertion-deletion (indels) signatures (ID1, ID2, ID5, and ID8) as major components of the mutational profiles (Fig 3C). We

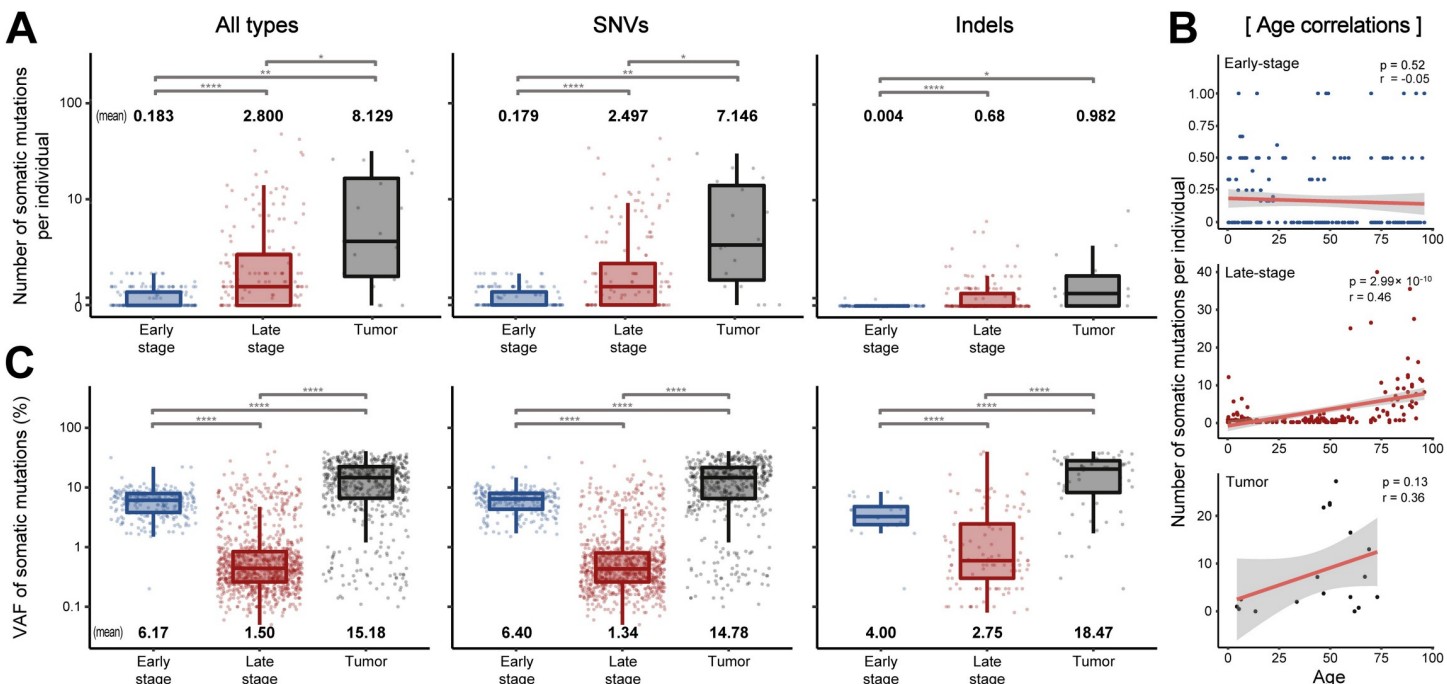

**Fig 2. Basic descriptive statistics for somatic mutations.** (**A**) The number of somatic mutations per individual in the early-stage, late-stage, and tumor mutation groups. (**B**) Correlation of age with somatic mutation counts in the groups. (**C**) Average VAFs between the three mutation groups.

noted that the relative contributions of the two well-known age-related mutational signatures (SBS1 and SBS5) were altered in early to late-stage SNVs (SBS1: 19% to 29%; SBS5: 78% to 49%).

## Differences in functional tolerance based on mutation timing

We identified functional differences between the early and late-stage mutations. In particular, early-stage mutations showed a lower ratio of non-synonymous to synonymous substitutions (quantified as the dN/dS ratio by maximum-likelihood methods) (0.79) than did late-stage mutations (0.94), tumor mutations (0.94), and common germ-line coding variants (0.90; gnomAD Exome) (Fig 3D), indicating a stronger negative selection [22]. Additionally, early-stage mutations were located in trinucleotides less frequently (2.1%) with atypical mutability [23, 24] than late-stage mutations (8.0%), tumor mutations (8.2%), and common germ-line coding variants (9.9%) (Fig 3E). Sites with atypical mutability are more highly mutated in cancer than is expected to occur randomly, indicating their functional significance and drive in cancer [24, 25]. Furthermore, genes that harbor early-stage mutations showed a lower probability of loss-of-function (LoF) intolerance (pLI score) [26] (Fig 3F), indicating that early-stage mutations are more enriched in LoF-tolerant genes.

## Organ-specific characteristics of late-stage mutations

Next, we investigated the characteristics of late-stage mutations, with a particular focus on diversity among different organs and cell types. The numbers of mutations varied substantially by organ, with a smaller number in the brain (0.67 per individual) and a higher number in the blood (9.23 per individual) relative to other peripheral organs (average: 1.12 per individual) (Fig 4A). However, the average value of VAFs was inversely proportional in these groups, with the highest number in the brain (7.32%) and the lowest in the blood (0.50%) (Fig 4B). The

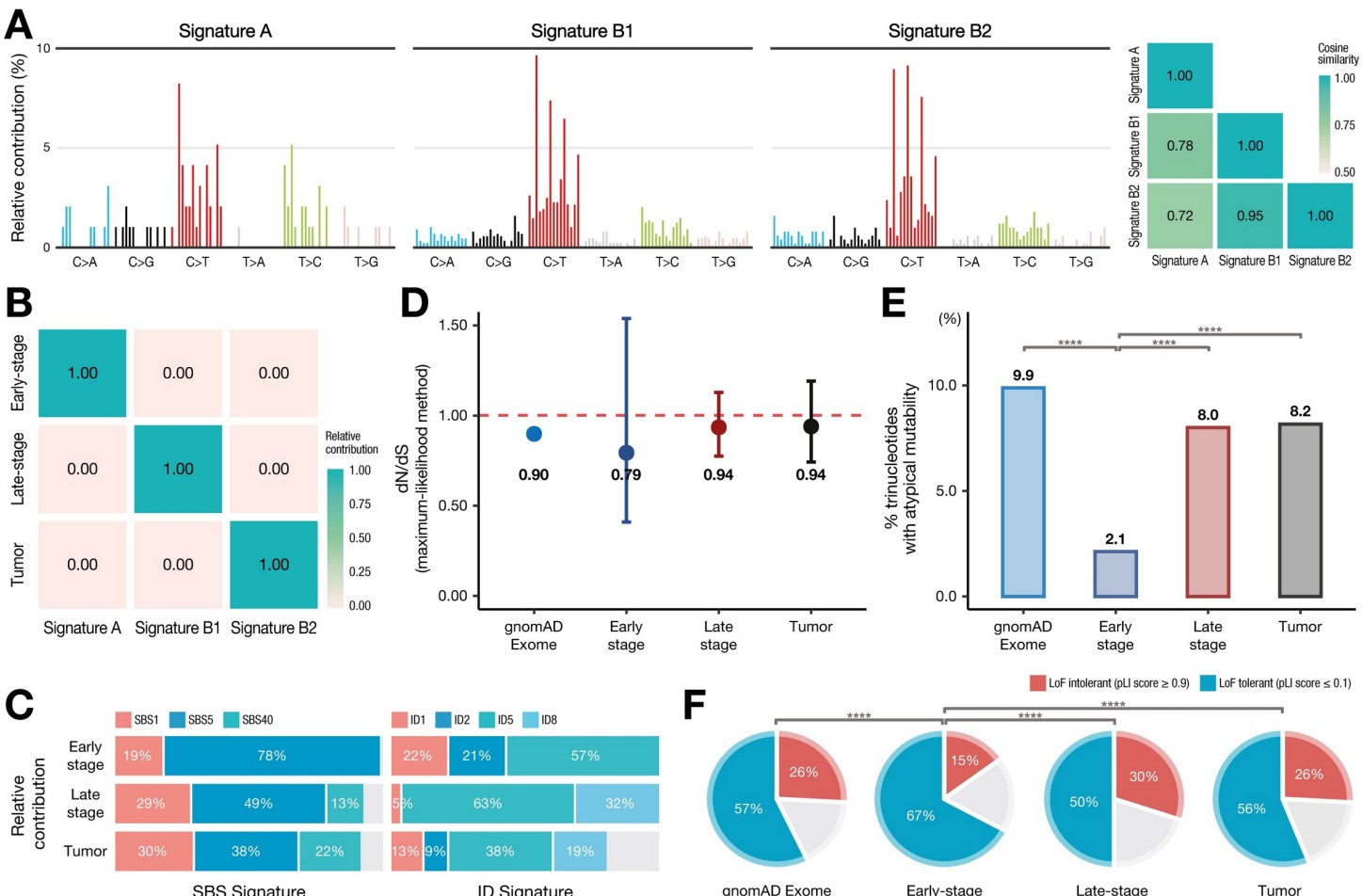

**Fig 3. Mutational profiles and functional analysis.** (**A**) *De novo* extraction of somatic mutations by non-negative matrix factorization. (**B**) Each group was classified according to the three signatures (A, B1, and B2). (**C**) The relative contribution of the common clock-like signatures (SBS1, SBS5, and SBS40 for single-base substitutions, and ID1, ID2, ID5, and ID8 for small insertion-deletions) from PCAWG signatures. Mutations from the three groups were extracted as three signatures by NMF. Mutational contributions were obtained by fitting the extracted signatures to the mutations in each group. (**D**) The dN/dS score ratios, (**E**) proportion of trinucleotides with atypical mutability, and (**F**) pLI score for gnomAD Exome and each group are shown.

number of late-stage mutations and the age of individuals showed a significant positive correlation in the blood ($r = 0.45$; $p = 2.99 \times 10^{-10}$; Fig 4C), and this phenotype has been observed in post-natal clonal hematopoiesis [27, 28].

Unsupervised hierarchical clustering of the three signatures (A, B1, and B2) in late-stage mutations showed that mutations in the brain primarily comprised signatures A (early-stage) and B2 (tumor), whereas blood mutations harbored signatures B1 (late-stage) and B2 (tumor) (Fig 4D).

We assessed the cell-type specificity of the somatic mutations in the brain by evaluating two brain samples. One (NLE-P-0150) contained early-stage mutations (5.47% VAFs) whereas the other (NLE-P-0225) contained five late-stage mutations (average: 8.00% VAF) (Fig 4E), each of which was sorted by fluorescence-activated nuclei sorting (FANS) to isolate three different cell types: neuronal (NeuN+), oligogenic (Olig2+), and others (negative). TASeq of the separated cell populations showed that both early and late-stage mutations were present in multiple cell lineages, but a large asymmetry in mutation frequencies was present among cell types with late-stage mutations (Fig 4E).

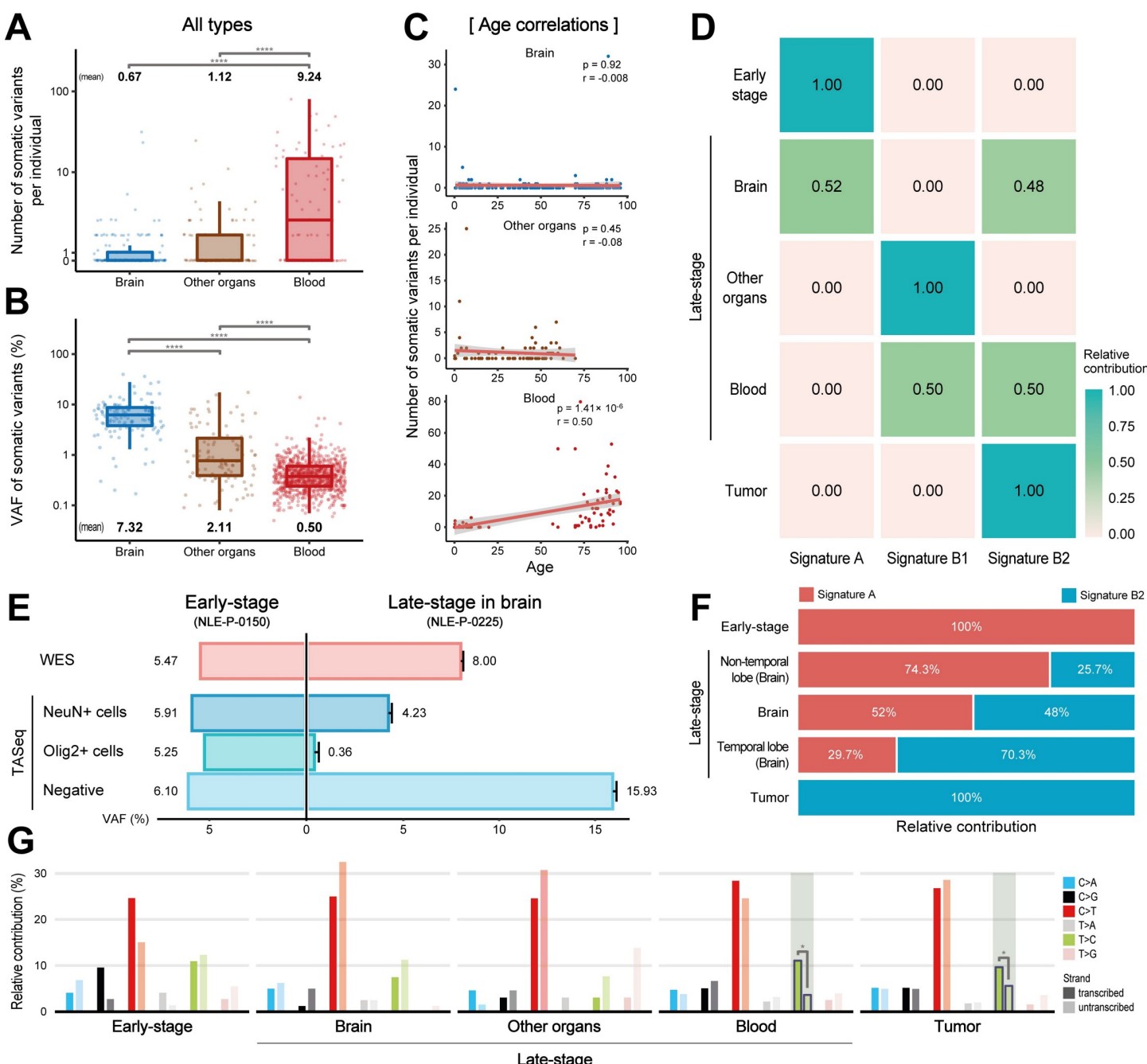

**Fig 4. Analysis of late-stage mutations based on the mutation source for the various organs (brain, blood, and other organs).** (**A**), (**B**) Number of mutations per individual and VAF distribution. (**C**) Age correlation with mutation counts. (**D**) Unsupervised hierarchical clustering of late-stage mutations. Late brain somatic mutations were fitted to signatures A and B2, whereas those in the blood were fitted to signatures B1 and B2. (**E**) The VAFs of three different cell types [neuronal (NeuN+), oligogenic (Olig2+), and others (negative)] for early and late-stage mutations in the brain. (**F**) Signature distribution of late brain somatic variants that were divided among temporal and non-temporal brain areas, or according to brain-disease status. (**G**) Mutational-strand asymmetry. Late-onset blood and tumor mutations are noted as having strand bias with T>C.

We further classified the late-stage brain mutations into temporal and non-temporal areas, and analyzed area-specific mutational signatures (Fig 4F). As reported previously, signatures A and B2 contributed to mutations in both areas; however, signature B2 had a higher contribution in the temporal lobe (70.3%) than non-temporal tissue (25.7%), indicating that the

characteristics of somatic mutations in the temporal lobe are more similar to those of tumor mutations.

Furthermore, late-stage mutations in the blood and tumor mutations showed enrichment for T>C mutations on transcribed strands (Fig 4G). Transcription-coupled repair occurs more frequently at higher transcription levels, and this bias is increased in actively replicating templates [29, 30]. We found that clonal expansion-derived somatic mutations were present in the blood, and were similar to those in tumors [31].

## Discussion

Based on a large-scale deep whole exome sequencing data analysis using a total of 498 matched sample pairs from 190 individuals, we provided a detailed picture of low-level but clone-forming somatic mutations, the associated mutation counts, and characteristics that are distinguished in time and space. We found that early-stage mutations, which arise before gastrulation and are shared by multiple organs, are lower in number and have a reduced functional impact than late-stage mutations that are restricted within a single organ. Moreover, we showed that late-stage mutations are associated with human mutational processes in the late-embryonic and post-natal developmental stages, but vary by organ, tissue, and cell lineages. In particular, late-stage mutations in the brain showed a bimodal distribution for developmental stages and the asymmetry of mutational features across brain-cell types and regions.

Using pairwise analysis of the brain and other matched organs, we identified mutational counts, variant fractions, and mutational processes with high confidence levels at each developmental or time stage. Our mutational numbers were roughly comparable to those from previous studies, which reported 0.53 and 3.15 shared and non-shared somatic mutations in the brain (numbers were normalized to a genomic size of 50 Mbp from whole-genome sequencing), respectively [16, 32]. We observed higher numbers of VAFs in the early stage (6.17 ± 3.32%) than in the late stages (1.50 ± 3.29%) of development, which is consistent with the hypothesis that somatic mutations that arise earlier have higher numbers of VAFs [3, 16]. VAFs for early-stage somatic mutations have been analyzed in several studies with different criteria for inclusion, and showed a diverse range (0.3%–55%) [3, 11, 32]. However, few studies have directly observed multi-organ-shared mutations using matched tissue sets from the same individuals; thus, our analysis provides a more realistic distribution of VAFs among shared or non-shared mutations in multiple-organs. Notably, VAFs of somatic indels in the early stages were lower than those of somatic SNVs (indels vs. SNVs: 4.00% vs. 6.40%), but higher in the late stages and tumors (2.75% vs. 1.34% in the late-stage; and 18.47% vs. 14.78% in tumors). The lower VAFs of indels, which indicates a lower cellular proportion and later occurrence, may be associated with lower tolerance to damaging mutations in the early-developmental phase.

In terms of mutagenic processes, we extracted three unique signatures that showed overall similarity but contained early stage, late stage, and tumor SNVs. This phenotype indicates that somatic mutations from different stages have different contexts. For mapping to aging-related COSMIC Mutational Signatures, the increased relative proportion of late-stage somatic mutations in SBS1 indicates the presence of active proliferation and clonal expansion during the late-embryonic and post-natal or aging periods [33, 34]. The greater contributions of the ID1 and ID2 signatures in early-stage indels, and ID5 and ID8 in late-stage indel signatures were consistent with a previous study, although the exact etiologies associated with most indel signatures remain unknown [15].

Early stage mutations were less tolerated functionally than late stage mutations. Because the early stages are critical for development, fatal mutations can have an embryonic-lethal

phenotype. Therefore, mutations are usually identified in functionally less essential regions. We found that early-stage mutations showed a lower ratio of dN/dS and pLI score, and were less frequently located in trinucleotides with atypical mutability. These functional analyses collectively indicated that strong selective pressure in the early embryonic stage [35, 36] affects overall mutation characteristics that are less damaging because of the rejection of functionally-deleterious mutations. Nevertheless, due to the small total number of early mutations, the signature analysis, and its interpretation require caution, and results from previous studies for early shared mutations have shown comparable results with our data [11].To define the characteristics of late-stage mutations, we assessed mutational profiles in organs and their cell type specificity. The numbers and variant fractions of mutations varied substantially by organ, and late-stage mutations showed fewer mutations in the brain (0.67 per individual), but high VAFs (7.32%). These results indicate that clonal somatic mutations in the brain occur relatively earlier but less frequently than those in the blood and other organs. Additional results for shared multiple lineages, including later-stage brain somatic mutations, showed a relatively early onset with a similar context. Mutational signature clustering with *de novo* signatures of late-stage mutations showed that mutations in the brain had early-stage and tumor mutation characteristics. The former result reflects the characteristic of low stemness and little or slow progression of division/differentiation in brain cells. Further analysis of late-stage brain mutations in the temporal lobe regions revealed that these were more similar to the signature B2 (tumor signature). These results suggest that late-stage somatic mutations in the brain show a bimodal occurrence during the early embryonic and post-natal periods, with a pattern similar to that in tumor mutations. We hypothesized that tumor-like mutational signatures in the temporal lobe may originate via neurogenesis (e.g., dentate gyrus) that confers clonal proliferation properties, as reported previously [37]. Asymmetric cell divisions resulting from early cellular bottlenecks for stochastic clonal selection contributed to an uneven variant fraction based on developmental timing [3, 38]. These findings suggest that the VAFs of clone-forming somatic mutations reflect the timing of the mutation, and the cell fitness and cell-type specificity for given somatic mutations.

Our study has several limitations. The study was performed using a limited range and number of tissues per individual, and the total number of mutations was small as identified by exome-level sequencing. Thus, finding sufficient SNPs or CNVs for analysis was difficult. We analyzed tissues from different donors with distinct germline or disease-susceptible backgrounds and lifestyles, thus making mutagenesis interpretation challenging. To address these issues, studies have been performed using sequencing for multiple tissues from the same donor [39]. Therefore, the analysis of multiple organs and different individuals can impair the identification of mutational process characteristics in individuals, but highlighting the results for common mutagenesis. Thus, although the common characteristics of each tissue were analyzed, the analysis of external and internal mutagenesis is difficult. An important consideration includes that somatic mutations should vary not only in cell types and tissues, but also in developmental stages, even in healthy individuals [5, 39, 40]. Therefore, it is difficult to have a representative set of normal tissues of other organs. To eliminate this bias, it is necessary to collect several types of tissues from all organs. Our study had a selection bias because age-mismatched "other organs" were a mixture of different organs, whereas brain tissues contained various locations with diverse ages, but diseased brain tissues were obtained from some patients. This can lead to erroneous results, and therefore, molecular properties were compared by subdividing tissues based on other organ compositions and types of brain disease (S1C and S1D Fig). Although there were differences in VAFs for each brain disease, all organs showed high VAFs compared to other organs or blood groups, as described above (S1C Fig). In other organ groups, the VAFs of the heart, liver, and lipoid tissues were 2.88, 1.56, and 0.96,

respectively, and there was a statistically significant difference between heart and liver samples (*p*-value = 0.03715, Wilcoxon rank sum test) (S1D Fig). Nevertheless, the difference in mean VAF values was within 1.32%, and the composition of other organs in this study had little effect on the conclusions. Another issue includes the uneven distribution of organs by age. For the brain, various age groups were included, whereas, for blood samples, there were many samples from patients who were under 20 and over 50 years old. For other organs, samples from patients in age groups under 50 years were included. Therefore, the number and VAFs for late-somatic mutations were not compared using age-matched tissue groups. We extracted data from patients aged 50–75 years, which was directly comparable in all three groups, and the molecular characteristics of late-stage mutations were maintained (e.g., fewer mutations and high VAF in the brain group, moderate mutations and VAF in other organ groups, and high mutation rates with low VAF in the blood group (S1E and S1F Fig)). Despite sampling limitations, our data was representative of late-stage somatic mutations.

Recent studies by Li et al. showed the presence of seven mutational signatures, including two age-related endogenous mutational signatures SBS1 and SBS5 [39]. The relative activities of SBS1 and SBS5 varied across tissues, particularly in the duodenum, colon, and rectum which showed a higher SBS1/SBS5 ratio than the bronchus, pancreas, esophagus, and liver [39]. In addition, exogenous mutational signatures were associated with organs and individuals [39]. We obtained more numbers of mutations in the late stage than those from the early stages. Nevertheless, it is difficult to conclude that such differences in numbers represent the mutation rate in the early and late developmental stages. This is because (i) it is difficult to determine the exact number of mutations due to differences in detectability (e.g., VAF) of clone-forming mutations in the early and late periods. (ii) The number of early and late mutations identified by our method was affected by the number and location of matched tissues within an individual, and it is difficult to conclude that these numbers are representative. (iii) It is difficult to identify an accurate mutation rate because the time difference between each stage and the rate of cell replication/differentiation differs from the time when these categories are established as early and late. Therefore, care is required in concluding that the mutation rate is high in late stages and that they are dominant.

Our body is remodeled by positively selected clones based on the survival of the fittest principle under given conditions, which can set the stage for cancer cell evolution even in normal aging, and the development of non-neoplastic diseases [5, 41]. The degree of remodeling differs substantially depending on the tissue type and the ecosystem, and external mutagens such as chronic inflammation, alcohol consumption, and tobacco smoking can affect the process [5]. In addition, microenvironmental decline, progressive decline of the competitive fitness of the tissue, and relatively low clearing of altered (dysfunctional) cells have been observed in aging tissues, some of which can be oncogenic [42]. In the case of the brain during somatic evolution, cellular division/differentiation is relatively slow or almost quiescent; thus, mutations that occur early in development are less frequent, but show high VAF levels. In the case of blood and other tissues, continuous differentiation proceeds, and if positive selection does not occur due to a special driver mutation, differentiation will result in a relatively low VAF level with a high number of mutations with similar cellular fitness.

The number of variants identified in this study was not large enough to seek read-based phasing or trace the developmental timing of somatic mutations based on WES, although these phenotypes have been investigated in previous large-scale genomic studies [16, 32, 38]. Somatic mutations are increasingly being identified via single-molecule sequencing, but the method is still in the early phases of development and prone to many errors due to culturing or amplification bias [43]. We found somatic mutations in clone-forming cells that would be detectable using WES, and mutational profiles of clone-forming somatic mutations can

account for functional effects, including cell fate, predominance, or natural selection. Overall, the well-defined characteristics of each mutational group and target tissue based on their developmental period can provide an accurate representation of currently-observable somatic mutations and an improved understanding of how these were generated.

## Materials and methods

### Ethics statement

The study was performed with written/verbal consent was obtained from the Participants, Patients or Parents according to the protocols approved by the Institutional Review Boards of Severance Hospital and Korea Advanced Institute of Science and Technology (KAIST), as well as the Committee on Human research. The developmental assessment was performed one month before surgery for child participants, and informed consent according to developmental age was obtained from the patient or patient/parent, then anonymized.

### Patient samples

The freshly frozen brain and peripheral samples acquired from 24 autism spectrum disorder (ASD) and five non-ASD cases from the National Institute of Child Health & Human Development (Bethesda, MD, USA) included various brain regions, such as the frontal, temporal, occipital, and cerebellar areas. Paired samples with other organs were derived from 13 ASD and five non-ASD cases, and brain samples were obtained from 11 ASD cases. The Stanley Medical Research Institute (Rockville, MD, USA) provided genomic DNA for brain tissue and other matched organs from 25 non-schizophrenic and 26 schizophrenic cases. Additionally, the institute provided genomic DNA for the brain and matched liver tissues from patients with major depressive disorders. Fresh frozen brain samples from patients with Alzheimer's disease (AD) were provided from the Netherlands Brain Bank (project number Lee-835) for 96 brain and matched blood samples for AD and non-demented control cases. Additionally, 15 samples from patients with AD and non-demented control cases were obtained from the Human Brain and Spinal Fluid Resource Center (West Los Angeles Healthcare Center, Los Angeles, CA, USA), which is sponsored by NINDS/NIMH (Bethesda, MD, USA), the National Multiple Sclerosis Society (Raleigh, NC, USA), and the US Department of Veterans Affairs (Bethesda, MD, USA). Fresh frozen samples of lumbosacral lipoma were obtained from the Severance Children's Hospital of Yonsei University College of Medicine (Seoul, Republic of Korea). Bone tissues of patients with non-syndromic craniosynostosis were provided from the Severance Hospital of Yonsei University College of Medicine. Individuals with refractory epilepsy, including focal cortical dysplasia and non-lesional epilepsy, and who had undergone epilepsy surgery, were enrolled through the Severance Children's Hospital of Yonsei University College of Medicine. Patients with glioblastoma and ganglioglioma were enrolled at the Severance Hospital of Yonsei University College of Medicine, and satisfied diagnostic criteria described in the 2016 World Health Organization Classification of Tumors of the Central Nervous System [17]. Tumor specimens were collected from the primary or recurrent tumor, and we evaluated tumor involvement via histological analysis, with normal brain or blood used as a control sample.

### Deep WES

Genomic DNA was extracted with either the QIAamp mini DNA kit (Qiagen, Hilden, Germany) from freshly frozen brain tissues or the Wizard genomic DNA purification kit (Promega, Madison, WI, USA) from blood according to manufacturer instructions. Each sample

was prepared according to Agilent library preparation protocols (Human All Exon 50 Mb kit; Agilent Technologies, Santa Clara, CA, USA). Libraries were paired-end sequenced on Illumina Hiseq 2000 and 2500 instruments (Illumina, San Diego, CA, USA) according to the manufacturer's instructions, and we used a confidence-mapping quality (mapping quality score $\geq$ 20; base quality score $\geq$ 20).

## Data processing and reciprocal somatic variant calling

We checked the quality of the raw sequencing reads using the FastQC [44] (v.0.11.7) software. The FASTQ-formatted reads from each sample that passed the quality checks were aligned to the human reference genome (build 38; NCBI, Bethesda, MD, USA) using the BWA-MEM [45] algorithm and converted into a BAM file. The initial BAM file was updated with the read groups, and duplicate information was excluded as the analysis progressed using Picard [46] and GATK [47]. Additionally, we performed local realignment and base-quality recalibration with GATK tools for each exome. The processed BAM files were used to analyze contamination between samples, with the probability of swapping assessed using the NGSCheckMate [48] software and cross-contamination tested using GATK tools. The Vecuum [49] software was used to check for vector contamination during library construction, and Depth of Coverage (GATK) was used to analyze sequencing depth. Processes that are not described in detail were performed based on the GATK best-practice pipeline.

Two or more tissue samples from each individual were analyzed using all-pairs testing. First, to detect somatic mutations in tissue "A", tissue "A" and "B" were used as a case and control, respectively, and further, tissue "A" was used as a control to identify somatic mutations in tissue "B". We used the somatic mutation-detection pipeline (paired mode) with sample pairs as inputs with a three somatic variant caller Mutect2 [18] and excluding the panel of normal creation (single nucleotide variants and small insertion-deletions), RePlow [19] (single nucleotide variants), and NeuSomatic [20] with the control of the false detection rate control performed by Varlociraptor [50] (small insertion-deletions).

## Systematic variant filtering and variant type discrimination

All mutations that met the following criteria were removed from the initial variant-calling results in the VCF format: oxoG-induced errors based on the method described by Costello et al. [51], common single-nucleotide polymorphisms by NCBI dbSNP [52] (build 153), segmental duplication and simple repeat regions according to the UCSC database [53], a mappability score under 0.8 by Umap [54], and the presence of off-target regions [55], whole genome data without exomes, and untranslated regions. After the removal of artifacts, somatic mutations showing "PASS" results for both Mutect2 and the filters of other callers (RePlow/ NeuSomatic) were considered organ-specific mutations. We determined allelic frequency concordance using a one-sample proportion test. The one-sample proportion test indicates individual differences based on a binomial proportion analysis, which is used for testing a hypothesized proportion compared to a given theoretical proportion of the population. Thus, we tested whether there is a difference between the individual's allelic frequency for somatic mutations relative to the population.

To determine whether multiple-organs shared mutations, variants that were present in matched sample pairs alone were included. In the filtering process, paired-mode somatic calling by Mutect2 was used. The probability of the candidates as multiple-organ shared mutations was identified using the combinations of the following identities: no correspondence to the error filter of other callers (RePlow/NeuSomatic), concordance of nucleotide substitution

type and genomic location in matched pair samples, and statistical validity using the one-sample proportion test.

Because differences in the number of samples analyzed can lead to biases, we corrected the data by dividing the number of detected mutations by the number of tissues.

## Validation sequencing using deep-targeted amplicon or Sanger sequencing

We performed validation sequencing by randomly selecting multi-organ shared and organ-specific mutations. For validation, we used deep-targeted amplicon sequencing or Sanger sequencing of PCR-amplified DNA, with the same genomic DNA used for deep WES. Primers for PCR amplification were designed using the Primer3 software (http://bioinfo.ut.ee/primer3-0.4.0/) [56]. Target regions were amplified by PCR using specific primer sets and high-fidelity PrimeSTAR GXL DNA polymerase (Takara, Shiga, Japan). Sanger sequencing was performed using BigDye Terminator reactions, and samples were loaded onto a 3730xl DNA analyzer (Applied Biosystems, San Francisco, CA, USA).

## Bioinformatics analysis

All somatic mutations with false positives excluded by validation sequencing were annotated using VEP [57] (v.99.0) with "-everything -plugin ExACpLI" options. The results were evaluated using an in-house script to analyze the descriptive statistics of the properties of the basic mutations, the effect of each gene, and potential correlations with patient demographics (age, disease, etc.). Non-negative matrix factorization-based novel signature extraction (NMF, 200 iterations) and transcriptional strand-bias analysis were performed using the Mutational Patterns program (R package) [58]. The signature and 96 variant context types were fitted to the clockwise Pan-Cancer Analysis of Whole Genomes (PCAWG) single-base substitution and small insertions and deletions signatures by deconstructSigs [59], Mutalisk [60] (date of use: March 2020), and YAPSA [61]. The maximum-likelihood dN/dS method was implemented using dNdScv (Wellcome Sanger Institute, Cambridge, UK) [22]. Mutability was calculated using NCBI MutaGene [23, 24] (v.0.9.1.0) which is distributed as a Python package.

## Nuclei extraction and fluorescence-activated nuclei sorting

Frozen brain samples were minced using pre-chilled razor blades and one or two drops of lysis buffer [0.2% Triton X-100, 1× protease inhibitor, and 1 mM DTT in 2% bovine serum albumin (BSA) in phosphate-buffered saline]. Lysis buffer (1 mL) was added to the homogenate and mixed by pipetting, after which the lysate was fixed in 1% paraformaldehyde at room temperature for 10 min. The fixed lysate was quenched with 0.125 M glycine at room temperature for 5 min. The homogenate was washed with suspension buffer (1 mM EDTA and 2% BSA), and filtered with a 40 μM cell strainer. The sample was incubated with anti-NeuN (mature neuronal marker; 1:1000 dilution, MAB377, Millipore) and anti-Olig2 (oligodendrocyte lineage marker; 1:500 dilution, ab10983, abcam) antibodies overnight at 4˚C, followed by washing with suspension buffer and staining with the secondary antibody for 1 h at 4˚C. After washing with suspension buffer, nuclei were passed through a 40 μM cell strainer and stained with 1 μg of 4′,6-diamidino-2-phenylindole. Nuclei used to isolate each cell type were analyzed and sorted using a MoFlo Astrios EQ cell sorter (Beckman Coulter, Brea, CA, USA). Nuclei pellets were centrifuged for 10 min at 1500 ×g and processed immediately for gDNA extraction using a QIAamp DNA micro kit (Qiagen) according to manufacturer instructions.

## Supporting information

**S1 Fig.** Late mutational analysis by mutagen in different organs (brain, blood, other organs) (A), (B) Age correlation with somatic mutation VAFs in late-stage variants and blood late-stage variants. (C) VAFs of late-stage somatic mutations in brains are classified by the diseases in individuals. (D) VAFs of late-stage somatic mutations in the heart, liver, and lipoid tissues. (E), (F) Number of mutations per individual and VAF distribution in individuals aged 50–75 years.
(TIF)

**S1 Table. Sample information enrolled in this study.**
(XLSX)

**S2 Table. List of post-filtered somatic variants.**
(XLSX)

## Acknowledgments

We thank the Netherlands Brain Bank (Lee-835) for Alzheimer's and unaffected control samples; the National Institute of Child Health & Human Development for providing Autism and unaffected control case samples; the Stanley Medical Research Institute for the brain and peripheral DNA of patients with schizophrenia, major depressive disorders, and unaffected control cases; Seoul National University Hospital, Seoul National University College of Medicine for providing lumbosacral lipoma and non-syndromic craniosynostosis samples; and Severance Hospital, Yonsei University College of Medicine for providing samples of brain tumor and epilepsy, which were supplied to J.H.L.

## Author Contributions

**Conceptualization:** Jeong Ho Lee, Sangwoo Kim.

**Data curation:** Ja Hye Kim, Shinwon Hwang, Hyeonju Son, Dongsun Kim.

**Formal analysis:** Ja Hye Kim, Shinwon Hwang, Hyeonju Son, Dongsun Kim, Yoo-Jin Ha, Jeong Ho Lee, Sangwoo Kim.

**Funding acquisition:** Hoon-Chul Kang, Jeong Ho Lee, Sangwoo Kim.

**Investigation:** Ja Hye Kim, Shinwon Hwang, Hyeonju Son, Dongsun Kim.

**Methodology:** Ja Hye Kim, Shinwon Hwang, Hyeonju Son, Dongsun Kim.

**Project administration:** Jeong Ho Lee, Sangwoo Kim.

**Resources:** Il Bin Kim, Myeong-Heui Kim, Nam Suk Sim, Dong Seok Kim, Junehawk Lee, Hoon-Chul Kang.

**Software:** Shinwon Hwang, Hyeonju Son, Dongsun Kim, Junehawk Lee.

**Supervision:** Jeong Ho Lee, Sangwoo Kim.

**Validation:** Ja Hye Kim.

**Visualization:** Ja Hye Kim, Shinwon Hwang, Hyeonju Son, Dongsun Kim.

**Writing – original draft:** Ja Hye Kim, Hyeonju Son, Jeong Ho Lee, Sangwoo Kim.

**Writing – review & editing:** Ja Hye Kim, Shinwon Hwang, Jeong Ho Lee, Sangwoo Kim.

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
