## [Decision Letter · Decision Letter 0]

1 Feb 2022

Dear Dr Kim,

Thank you very much for submitting your Research Article entitled 'Multi-organ analysis of low-level somatic mosaicism reveals stage- and tissue-specific mutational features in human development' to PLOS Genetics.

The manuscript was fully evaluated at the editorial level and by independent peer reviewers. The reviewers appreciated the attention to an important problem, but raised some substantial concerns about the current manuscript. Based on the reviews, we will not be able to accept this version of the manuscript, but we would be willing to review a much-revised version. We cannot, of course, promise publication at that time.

In addition to the reviewers' concerns, I would request that in a revision you also carefully evaluate terminology that refers to mutations as pre- or post-gastrulation, since on my read it was not always clear that certain mutations were present in all three germ layers (especially when dealing with limited distribution of tissue sources and the possibility that tissues contain a mixture of germ layer origin.

If you decide to revise the manuscript for further consideration at PLOS Genetics, please aim to resubmit within the next 60 days, unless it will take extra time to address the concerns of the reviewers, in which case we would appreciate an expected resubmission date by email to plosgenetics@plos.org.

[LINK]

We are sorry that we cannot be more positive about your manuscript at this stage. Please do not hesitate to contact us if you have any concerns or questions.

Yours sincerely,

Marshall S. Horwitz, M.D., Ph.D.

Consulting Editor - PLoS Genetics

PLOS Genetics

Gregory Barsh

Editor-in-Chief

PLOS Genetics

Reviewer's Responses to Questions

**Comments to the Authors:**

Reviewer #1: The authors present an interesting study that leverages whole-exome sequencing (WES) data from multiple tissues from a large number of individuals (190) to gain insight into the nature of somatic mutations that occur in early and late embryonic development. The authors classify somatic mutations as early if they are shared by two different tissues and late if they are not. In addition, the authors also sequenced a subset of brain tumors for comparison. The main finding is that early mutations have higher allele frequency and lower functional impact than late mutations as revealed by multiple genetic metrics (lower dN/dS, location in trinucleotides with atypical mutability, and location in LoF-tolerant genes). In addition, novel mutational signatures and their contributions to different brain regions are described.

While the approach is interesting and there is a high need in the field for this sort of comparative analysis, the article has some important caveats, described below.

1. The study is far from being a systemic and comprehensive analysis of low-level somatic mutations, as claimed in the abstract. Only 3 tissue types (brain, blood, and liver) are analyzed in a large number (60+) of individuals. Authors claim multi-organ sequencing but for most individuals only two tissue types are analyzed: either brain and blood, or brain and liver. The authors should tone down the claim of multi-organ analysis since the study is essentially on brain tissue, just using different control tissues.

2. The article is difficult to read because some key points are not clearly explained. For example, in the 19 individuals with brain tumors, the brain tissue analyzed was from tumor or from normal? That is unclear in figure 1 and it is not clearly explained in Material and Methods. The text only mentions: “If the source of the sample was related to a brain tumor, it was separately regarded as tumor mutations.” Please clarify the origin of samples in individuals with cancer and whether brain tumor tissue was used for the case-control comparisons explained in Fig. 1A. If it was used, this could be a source of confounding because in individuals with cancer, shared mutations tumor-blood could be due to ctDNA or cancer cells disseminated in the blood stream.

3. The legend for Fig. 1A needs more explanation, especially regarding what samples are considered cases vs controls and the one-sample proportion test. What are the tests for organ specificity?

4. When quantifying the number of early and late-stage somatic mutations per individual, how do the authors correct for the fact that different number of samples and different tissues were analyzed in different individuals? The article might benefit from removing the ‘other tissues’ and ‘heart’ subgroups since there are not many samples in those subgroups and will make the comparisons more similar across individuals. The different sample types used for different individuals is a concern.

5. It is unclear how the mutational signature analysis might be relevant when the late-stage mutations come from a mixture of different tissues including brain, heart, liver, blood and others. Different tissues harbor different mutational tissues, please see PMID: 34433965 and 34433962. In addition, the number of early-stage mutations is small (94) for robust mutational signature analysis.

6. The comparison of number and VAF of late-stage somatic mutations by tissue type should take into consideration the fact that the 3 groups of tissues are not age-matched. The is no data for older individuals for ‘other organs’.

7. The brain tumors included in the study are glioblastoma and ganglioglioma. Glioblastoma is an adult tumor and ganglioglioma is a child tumor. Is it possible that the unique tumor mutational signature identified is the result of the mixture of mutations derived from these two very different brain tumors? If the ganglioglioma tumors were removed, would the mutations observed resemble to the landscape of mutations (driver genes) and mutational signatures reported for glioblastoma?

8. Very low resolution of figures makes it impossible to read some of the content.

Reviewer #2: Hyeonju Son and associates analyzed mosacism for SNVs and indels by means of deep whole-exome sequencing in 498 tissues from 190 individuals. Their findings were consistent with those of previous studies: somatic mutations with apparent early embryonic origin were fewer in number but had higher variant allele frequencies than variants that appear to have arisen later in life. Mutational signatures among early embryonic and later-arising variants were similar but differed somewhat.

This research is well described, and the paper is generally well written, although there are occassional errors of English usage. There are a few issues that require attention:

Line 233: "Early-stage mutations were not functionally tolerated than those of late-stage mutations." What does this mean? Are some words left out?

Discussion: The authors should discuss the limitations of this study (limited range of tissues studied, limited numbers of tissues per patient, inability to detect some kinds of variants such as CNVs, etc.).

Line 292: "Subjects with refractory epilepsy, including focal cortical dysplasia and non-lesional epilepsy, and who had undergone epilepsy surgery were enrolled through the Severance Children’s Hospital of Yonsei University College of Medicine." If these samples were obtained because of regional brain tissue excision for intractable epilepsy, could the mutations found have been responsible for the seizures? If so, might the frequency of variants and/or the VAF be biased to higher values in these samples? The same question could be raised in any other group of samples obtained by surgery to remove pathological tissue. This is an issue that the authors should discuss.

Reviewer #3: Throughout its lifetime, human body accumulates mutations, which despite affecting multiple body parts rarely cause a pathology. To provide a better grasp of spatiotemporal characteristics of this genetic mosaicism, the Authors of the manuscript performed deep whole exome sequencing of human tissues comparing the pattern of clone-forming mutations in various organs. In its focus on differences between early (pre-gastrulation) and late (post-gastrulation) mutations, this well-designed study addresses an interesting, and largely unexplored problem of developmental shifts in somatic mutagenesis and makes up an important contribution to current studies about somatic mosaicism.

The results include the finding that most somatic mutations are produced after gastrulation and that diversity of late-appearing mutations is higher than of the early ones. However, it is not clear if the observed numerical differences between early and late clone forming mutations reflect distinct mutation rates during these developmental stages. More details should be also provided to elucidate whether differences in mutational signatures across tissues stem from the action of distinct (internal and external) mutagens in these sites. They Authors should refer to recent studies by Li et al., Nature 2021, 597 (7876): 398-403, who also considered the problem of genetic mosaicism and compared the mutational patter across tissues.

I see the value of the present manuscript in its potential to contribute to ongoing studies of clonal evolution and suggest that the Authors elaborate on how their work contributes to recent studies on this subject (Kakiuchi and Ogawa, Nat Rev Cancer 2021, 21(4):239-256; Swiatczak, Cell Mol Life Sci, 2021 78(21-22):6797-6806; Laconi et al. 2020, Br J Cancer 2020, 122(7):943-952). For example, the Authors hint that variant allele frequency (VAF) can be used as a measure of a degree of clonal expansion of mutated cells in various tissues but draw no conclusions pertaining to the question of somatic evolution based on the observed differences between VAF in the brain, blood and other tissues.

All in all, the present manuscript provides an important contribution to cutting edge research on genetic mosaicism, which is getting a lot of attention now, but extending the present research to incorporate the above questions could help to shed light on the mechanisms underlying the observed mutational profiles.

**Have all data underlying the figures and results presented in the manuscript been provided?**

Reviewer #1: Yes

Reviewer #2: Yes

Reviewer #3: Yes

PLOS authors have the option to publish the peer review history of their article (what does this mean?). If published, this will include your full peer review and any attached files.

Reviewer #1: No

Reviewer #2: No

Reviewer #3: No

---

## [Decision Letter · Decision Letter 1]

13 Jul 2022

Dear Dr Kim,

Thank you for responding to the reviews. All three reviewers feel the manuscript is significantly improved. However, they have continued to identify outstanding concerns that prevent publication at this time, including, in addition to scientific issues identified by Reviewer 1, a need for revision of English usage. We hope that are able to address these concerns and revise the manuscript accordingly.

[LINK]

Yours sincerely,

Marshall S. Horwitz, M.D., Ph.D.

Consulting Editor - PLoS Genetics

PLOS Genetics

Gregory Barsh

Editor-in-Chief

PLOS Genetics

Reviewer's Responses to Questions

**Comments to the Authors:**

Reviewer #1: The authors have made a substantial effort to address my questions and concerns in the responses to reviewers and have added a very extensive paragraph about the limitations of the study in the Discussion section, which I appreciate. However, I believe the manuscript still lacks some of the necessary amendments.

The first one relates to the problem of different ages for the different sample types, which is very obvious from Fig. 4C. The authors provided an analysis restricted to patients aged 50 to 75 in the responses to reviewers. I believe this data should be discussed in the text and included as a supplementary figure.

The second issue relates to the selection of tissues for the study, which was done based on convenience. This leads to the “other organs” being a random mixture of tissues, which is still not well explained in the article, and to the brain tissue including a large proportion of samples from diseased tissue. The article would greatly benefit from more clarity on the types of samples included and from additional plots (which could be included as supplementary data) in which the authors further analyze the data to demonstrate that the choice of samples is not biasing the results. For example, in response to Reviewer #2, the authors indicate that the VAF of disease tissues is not different from the normal. All those comparisons should be clearly presented with dot plots in the article. Also, the authors have included this sentence in results: “VAF of late-stage mutations was inversely correlated with age, probably due to the accumulation of late-somatic mutations in blood with a low allelic fraction with aging”. This plot should be shown for all late-stage mutations and also for blood mutations only.

Finally, the article still requires attention to the explanation of the figures and methods. What are the numbers in the bars of Fig. 1A? what do red and blue colors indicate? In Fig. 3B, how are the relative contributions calculated? In Methods, there are some sentences that are hard to understand. For example: “In the filtering process of paired-mode somatic calling using Mutect2, “normal artifact” is corresponding to it; variants are non-germline mutation, differ from reference sequence, and contained in matched pair samples. “ Overall, the article would benefit from careful reading to improve some language issues and to make some of the explanations in methods and results clearer.

Reviewer #2: Hyeonju Son and associates have revised and substantially improved their analysis of somatic mosaicism using deep exome sequencing. They have adequately addressed all of my previous major concerns.

There are still numerous errors of English grammar and word usage; this paper would benefit greatly from copy editing for English usage.

Reviewer #3: The manuscript has been much improved in terms of clarity as the Authors have identified the origin of tumor samples and provided essential details about the mutation calling methods. They also brought the importance of their findings into light by relating their research to other somatic mutagenesis and clonal evolution studies.

**Have all data underlying the figures and results presented in the manuscript been provided?**

Reviewer #1: Yes

Reviewer #2: Yes

Reviewer #3: Yes

PLOS authors have the option to publish the peer review history of their article (what does this mean?). If published, this will include your full peer review and any attached files.

Reviewer #1: No

Reviewer #2: No

Reviewer #3: No

---

## [Decision Letter · Decision Letter 2]

31 Aug 2022

Dear Dr Kim,

We are pleased to inform you that your manuscript entitled "Analysis of low-level somatic mosaicism reveals stage and tissue-specific mutational features in human development" has been editorially accepted for publication in PLOS Genetics. Congratulations!

Yours sincerely,

Marshall S. Horwitz, M.D., Ph.D.

Consulting Editor - PLoS Genetics

PLOS Genetics

Gregory Barsh

Editor-in-Chief

PLOS Genetics

Comments from the reviewers (if applicable):

Reviewer's Responses to Questions

**Comments to the Authors:**

Reviewer #1: The authors have made a substantial effort to address my concerns, including the addition of a supplementary figure that increases confidence on the results. I find a bit unusual, however, that panels C to F of Supplementary Figure 1 are not presented in Results, but in Discussion. That said, I believe that the extra figures have improved the manuscript. Clarity is still an issue and the manuscript would greatly benefit from language editing.

**Have all data underlying the figures and results presented in the manuscript been provided?**

Reviewer #1: Yes

PLOS authors have the option to publish the peer review history of their article (what does this mean?). If published, this will include your full peer review and any attached files.

Reviewer #1: No

**Data Deposition**

http://datadryad.org/submit?journalID=pgenetics&manu=PGENETICS-D-21-01628R2

**Press Queries**

---

## [Editor Report · Acceptance letter]

13 Sep 2022

PGENETICS-D-21-01628R2 

Analysis of low-level somatic mosaicism reveals stage and tissue-specific mutational features in human development 

Dear Dr Kim, 

We are pleased to inform you that your manuscript entitled "Analysis of low-level somatic mosaicism reveals stage and tissue-specific mutational features in human development" has been formally accepted for publication in PLOS Genetics! Your manuscript is now with our production department and you will be notified of the publication date in due course.

With kind regards,

Anita Estes

PLOS Genetics

On behalf of:
